# Impacts of Second Home and Visiting Friends and Relatives Tourism on Migration: A Conceptual Framework

Sonia Ferrari

Department of Business and Legal Science, University of Calabria, 87036 Cosenza, Italy; sonia.ferrari@unical.it

**Abstract:** What is the thread that unites tourism and migration? A review of existing literature suggests two forms of tourism linked to migration: visiting friends and relatives and second home tourism. Tourism related to visiting friends and relatives can be stimulated by migratory movements, and, in turn, gives rise to new migrations. Second home ownership, however, serves as the main connection between tourism and migration, promoting tourism that generates from or relates to current and past migrations. This exploratory study is based on a qualitative systematic literature review and focuses on the characteristics of second home and visiting friends and relatives-related tourism, and migration, in order to clarify little studied linkages among them that can affect tourism-related development. The study shows that many of the migration-led tourism segments reflect factors that may promote sustainable development.

**Keywords:** visiting friends and relatives; roots tourism; second home; residential tourism; place attachment; migration

## 1. Introduction

Development problems do not affect only third world countries. Even in European countries and other Western economies there are isolated and less developed inland areas and small towns and villages that often suffer from unemployment and depopulation. For these areas, tourism could be one of the driving productive sectors, especially if it is developed in a sustainable way and along a path that does not have a negative impact on the local socio-cultural and economic fabric; on the contrary, it can promote its development [1–6].

However, the study of tourism as a development tool in isolated and economically backward areas needs the adoption of a prospective view. Tourism must be analysed as a catalyst for the development of the local and host community. In this sense, the study must be assessed with reference to other phenomena and economic activities that may or may not favour the positive effects of tourism [7–9].

Therefore, understanding the strength of the link between tourism and the local community, especially regarding the type of connection, is important. It helps to reach a deeper knowledge about the attitude of residents concerning this economic activity and its potential growth, the interest of the community in the phenomenon, and the qualitative and quantitative aspects of its potential impact on the local economy and quality of life. The link between tourism and the local community is frequently affected by ancient family roots, place attachment and other feelings and emotions towards the motherland concerning emigrants and their descendants. Hence, to consider tourism in conjunction with past and future migrations makes it possible to better appreciate its many effects at the local level as well as their evolution. This could be a useful approach for understanding the impact of tourism in terms of sustainability, i.e., how it affects not only economic variables, for example the levels of a community's income and unemployment, but also the social and cultural aspects, such as local entrepreneurship, civic pride, place image, social wellbeing and so on [10,11].

The article focuses on the characteristics of second home (SH) tourism, visiting friends and relatives (VFR), and migration, with the aim of clarifying the linkages between tourism, mobility, and migration flows, which significantly affect tourism linked development and will transform this industry and its impact in the future. The main objective of this research is to expand on the theorical foundation of the issue and provide an interdisciplinary and innovative overview of the interconnections between these two important and international mobility elements by examining what factors represent a common thread in this field of research.

A review of existing literature and an analysis of several macro-phenomena affecting consumption, lifestyle, migration, and the tourism industry today suggest that two interrelated elements serve as linkages between tourism, its effect in terms of local development and migratory movements: the desire to visit friends and relatives and SH ownership [12–29]. The research question that arises is: What effect will these two factors have as real or potential trigger elements linking tourism and migration? The research findings show that some tourism segments, namely VFR and SH, are seen as forms of migration-led tourism. This study determines that, on the one hand, VFR tourism is partly stimulated by migratory movements, and in turn triggers migrations as well; on the other hand, SH tourism, which is recognized in the literature as the main link between tourism and migration, involves flows of multiple visitors directly generated from or related to past and current forms of migration.

Analysing the interactions between tourism and migration through this new perspective explores aspects of these interactions that have frequently been neglected in extant literature and identifies areas that require additional research. This will allow new elements to be explored in future research and facilitate theory development. Moreover, it may encourage a more comprehensive approach to the study of phenomena that have been studied but which—in recent decades—have been changing rapidly, and therefore need different analysis tools and a new research perspective.

*Methodology*

This paper is based on a qualitative systematic literature review [30], aimed at systematizing the results of previous studies and highlighting new perspectives and research paths. The search was made using the Google Scholar database with no time restrictions. Initially broad keywords, such as 'migration and tourism', 'migration, tourism and mobility', were used. Later it was made recourse to more specific keywords ('VFR and migration', 'place attachment and tourism', etc.). For every paper the preliminary relevance was determined by title. The number of potential studies that were found was rather high (more than 400). Later, a qualitative selection of the articles was made considering the scope of this study. The abstracts have been read to understand if they were interesting and useful for this research. After this selection, the number of relevant studies was 153. Another selection was made on the basis of the quality and eligibility of each study. Some of them (3) were impossible to find, others (16) use a no clear methodological process. Finally, 134 papers were selected. Subsequently, another 27 articles were found through backward and forward search. A total of 161 articles were consulted for this study.

After an assessment of the results of the research, information was extracted from articles to aggregate and integrate existing literature on the research theme. The aim was to extend existing literature but also to give birth to new constructs through a thematic synthesis that could answer the main research question [31–33]. Data analysis followed an inductive approach. Data have been managed by hand through a thematic content analysis, identifying four main themes: tourism and migration in general; migration-led tourism and tourism-led migration; second homes and migration; and visiting friends and relatives and migration.

## 2. Tourism and Migration

Today, many socio-cultural and economic trends affect tourism and migration. With growing mobility in an increasingly globalized world, in which travel and communication are easier, faster, and more affordable than ever—numerous innovations are transforming tourism and migration [34]. In 2020, the number of international migrants was estimated at 280.6 million, constituting 3.6 per cent of the world's population. Two-thirds are labour migrants [35]. The number of international migrants has grown by 83 per cent since 2000. The major flows are represented by migrant workers (169 million in 2019) [36]. In 2019, 74 per cent of international migrants were working age (20–64 years), while only 12 per cent were age 65 or older; this percentage rose to 15 per cent in the more developed regions [37]. Such migration trends will continue, due to the increasing number of pensioners in Europe and due to the availability of information and communication technologies, which stimulate and encourage distance communication and smart working [38–40]. In 2019, the number of international tourists reached 1.5 billion, representing the tenth consecutive year of sustained growth since 2009. During 2020, total international tourist arrivals decreased by 72 per cent because of the COVID-19 pandemic (period January–October 2020) [41]. Of course, the pandemic makes future tourism trajectories unknown.

Tourism and migration are important international phenomena that are interdependent and characterized by similar dynamics [40,42]. The term 'migration' is an intricate and manifold concept. It has been defined in spatial terms as a 'movement across the boundary of an areal unit' [43] (p. 34); it has also been temporally defined: 'it is generally agreed that there will be some permanence to a move described as a migration' [43] (p. 35). The reasons to migrate vary along a continuum from economic to lifestyle motivations, and frequently there are multiple reasons [44,45]. The main reasons are work, family, and study.

Even the concept of 'tourism' is difficult to define [46,47]. An important and frequently used definition is from the UNWTO [48] (p. 5): 'Tourism comprises the activities of persons travelling to and staying in places outside their usual environment for not more than one consecutive year for leisure, business and other purposes.' Tourism gives rise to migratory flows—'tourism-led migration'—with migration also leading to 'migration-led tourism' [27,34,49]. Migration promotes tourism both from and to one's home country to a migrant's new place of residence. On the one hand, migrants, who usually maintain strong emotional and social ties with their mother country, enjoy vacations to see friends and relatives in their homeland [50,51]. Moreover, friends and relatives visit migrants in their new places of residence [29].

### 2.1. Migration-Led Tourism (MLT)

Tourism and migration are interrelated forms of mobility, with increased and intense interaction between them documented in recent years [52–54]. In general, the link between tourism and migration has been neglected in academic studies [27]. However, in the main tourism segments, the underlying motivations for travel are often permeated by elements linked to migration [50], as seen, for example, in the case of VFR tourism. This is a form of travel 'involving a visit whereby either (or both) the purpose of the trip or the type of accommodation involves visiting friends and/or relatives' [55] (p. 4). Many VFR tourists are permanent migrants that have maintained close relationships with their mother country and, from time to time, visit it [19]. This segment also includes family members and friends visiting migrants [12,13,17,18,27]. However, due to the multiple motivations of VFR travellers, they are sometimes considered parts of other segments, and they have been called 'hidden VFR travellers' [13]. In some destinations, VFR is the main form of tourism [52]. It can reach 70 per cent or more of arrivals in some countries, especially in Central America [50]. Furthermore, the VFR segment is continuously growing [56].

Migration can give rise to diverse types of travel [57]. These include the journeys of emigrants to their homeland and visits by family members and friends to emigrants' new homes. To date, the latter travelling pattern has been understudied [14,29]. In addition, emigrants may travel to other places to meet members of their extended communities.

Other emigrants spend holidays at destinations where people of similar ethnic origin live. Furthermore, emigrants may engage in roots tourism, which occurs when migrants and their descendants visit their countries of origin. All five types of migration are part of, or overlap with, the VFR segment. Trips motivated by VFR can impact future tourist flows and promote migration. VFR tourism can also promote visits to the homelands of other emigrants due to conversations about their home countries [28]. Thus, migration and VFR tourism are strongly interconnected phenomena: the former represents a pre-condition for VFR tourism, and the latter can be classified as a form of MLT [21].

### 2.2. Tourism-Led Migration (TLM)

Roots tourism (RT) is a fast-growing segment of heritage tourism. The main motivation of roots tourists is the desire to discover their roots and to recollect with their family's land of origin [14,58]. RT has the potential to be a major tourist segment in many destinations, especially in those countries marked by histories of consistent migratory flows. It is closely interrelated with not only past and present day migration but also different types of tourist arrivals, including those generated by VFR tourism and by the possession of SHs [59].

Empirical attention to this international phenomenon is rather recent, despite the great development potential of RT and the advantages it can generate in terms of sustainability, seasonality reduction, longer stays, growing interest in smaller destinations, and propensities for the purchase and the promotion abroad of local products [60,61]. Not only are there only a few studies on the subject, with official statistics missing, but investments in institutional initiatives aimed at incentivizing this segment are scarce as well [23]. In this respect, some countries are exceptions, as they have promoted RT through specific initiatives: Ireland, Scotland, Germany, Poland, and recently also Italy [59,62].

RT and VFR tourism segments do not always overlap, as the primary reason for RT may not be to visit family and friends. However, first and second immigrant generations generally manage to maintain fairly strong ties with their families in their countries of origin, with the desire to see them again almost always being one of their main travel motivations [59,63]. In such situations, RT could be considered a sub-segment of the wider VFR segment [64] (p. 855). Both RT and VFR can have numerous travel motivations [65]. As such, a distinction must be made between VFR as the main motivation for the trip and VFR as one of many activities carried out during a holiday experience [66]. As such, there remains a need for further studies on these two segments and their interrelationships.

Migration and RT are also linked to forms of 'event tourism' [67–69]. Emigrants often return to their mother countries to take part in cultural and religious events. In the cases of family events, such as weddings, baptisms, or funerals, there is an overlap with the VFR tourism segment.

Finally, the important segment of 'migrant tourism workers' must not be forgotten [70,71]. They are skilled and semi-skilled workers whose casual lifestyle and search for employment push them to move from one tourist destination to another. They are part of the 'tourism-labour migration,' which originates from the need for personnel from tourism enterprises that is met by young people looking for employment, leisure activities, and travel [72,73].

## 3. Second Homes, Place Attachment, and Migration-Led Tourism

SH or residential tourism has only recently been identified as a link between tourism and migration [74,75]. SH tourism comprises travellers who own a second property outside their place of residence and use it for leisure and recreational purposes [17,22,76–78]. SH owners are called 'permanent' or 'marginal' tourists [53,79]. Some scholars go as far as to say that they should not be considered tourists but rather 'new residents' [79,80] or 'semi-permanent migrants' [74,81].

SHs include different types of accommodations (e.g., holiday homes, summer houses, weekend houses), with the diversity garnering ambiguous treatment in the existing literature [75,82]. Today, the study of this form of tourism does not concern only Western

countries as SH locales but many other regions as well [83–87]. In fact, globalization, the transition from industrial to post-industrial society, and an increase (since the mid-1990s) in the number of homes owned abroad have led to a renewed interest in this form of tourism that had almost disappeared as an object of international academic research in the 1980s [22]. In many European countries, including northern Europe, the United Kingdom, Germany, Italy, France, and the Czech Republic, but also outside Europe in the United States, Canada, South Africa, Australia and other countries, there is a widespread tradition of owning SHs [88,89].

SHs served as ways to escape urban life for some months during the year [90]. Their main appeal was, and still is, their reversal of everyday life; however, other reasons also exist for buying an SH, with motivations that are complex and related to various factors, including characteristics of the area, presence of recreational opportunities, distance from one's place of residence, accessibility, local lifestyle, family ties with the territory, personal plans for retirement, a desire to live closer to the natural environment, a desire to return 'home,' improved lifestyle, investment motivations, and available amenities [84,91,92]. SHs are often linked to family history and/or childhood or youth. A significant and understudied aspect of SH tourism is the role of 'place attachment' as a motivating factor [93,94]. This refers to a strong connection with a location or house that garners remarkable loyalty when making holiday-related choices. Emigrants frequently hold this feeling with respect to their homeland and/or home of origin [95,96].

The ancestral bond of migrants and roots tourists with their homeland is strong, serving as the basis for the desire to (re)integrate with their homeland communities [97–100]. Frequently these communities do not perceive some returning migrants as tourists, and they do not see themselves as tourists but as a part of the community that has moved away [95,101,102]. Moreover, such a bond creates emotional conditions that increase the temporal frame of tourist visits from short-term holidays to extended stays and longer-term investments, including the purchase of properties, such as the house where they or their ancestors lived [103,104]. For this reason, RT is intertwined with residential tourism.

For emigrants and their descendants, the SH represents a way to maintain ties with their homeland and their relatives and friends, relive childhood memories, and increase closeness with their heritage [14,24,25]. In such cases, the 'home' acquires a particular and personal meaning, symbolizing the desire of emigrants to maintain a connect with a place that has affectionate attachment for them but where they cannot or do not wish to reside. The concept of place attachment has been largely neglected in research on SHs [83,96].

The positive attitudes of roots tourists towards their homeland and their strong ties to its culture and lifestyle reflect the respect of this tourist segment for the socio-cultural integrity of the host community and its cultural values and traditions; it positively influences the community's socio-cultural fabric. For this reason, it represents a sustainable form of tourism, whose development could be a tool to empower host communities by enhancing local identity, cultural values, integrity, and quality of life [105].

## 4. SH Tourism and Mobility

As illustrated above, there is an interdependence between temporary and permanent mobility, with SHs regarded as a linkage between them [106]. Nevertheless, the relationship between migration and SH ownership and residential tourism has been neglected by existing literature [26,53,107].

An SH is often purchased with a view to a future change of residence, especially by people who want to improve their lives and take advantage of cost and quality of life differentials compared to their usual place of residence. These transfers give rise to retirement, investment, and lifestyle migration, all of which represent privileged forms of migration and give birth to long-term tourism flows [108–110]. Such migratory patterns originate largely from Western countries and promote significant flows of SH tourists; they are primarily composed of people, especially pensioners, who settle, permanently or almost permanently, in a new place, sometimes in their family's country of origin. European

pensioners and lifestyle and investment migrants are increasingly deciding to live in foreign countries within Europe, such as Spain, Italy, and France, but also for instance Turkey, Portugal, and Bulgaria [111,112]. They are frequently attracted by countries offering good climate and pleasant lifestyle as well as, in some cases, tax benefits, economic advantages, and political stability [57,113]. This phenomenon is encouraged by the fiscal policies of some governments.

In future years, while the demand for SHs will continue to grow due to phenomena such as retirement and lifestyle migration [57], reduced interest in SHs among younger generations is also expected. Younger generations are more eager to enjoy experiences through shared access rather than owning goods [114–117]. Consequently, for them the temporary possession of goods, facilitated by web platforms, is becoming more popular than ownership [114,117].

These trends are accelerated by the development of the sharing economy and, in particular, the success of accommodation sharing [118–120]. Accommodation sharing is an increasingly popular business and consumer model based on the sharing of under-utilized housing through digital platforms, which are peer-paid or free of charge. It is completely revolutionizing the marketplace for home rentals, property investments, SHs, social housing, and tourist accommodations.

## 5. SH Connections with Various Forms of Migration

An SH can be considered a link between tourism and migration in many forms. SH ownership is connected to various migratory phenomena, such as retirement, amenity, lifestyle, and investment migration [62]. These are increasingly new forms of migration that can, in part, be considered types of tourism. They are not generated from labour or economic needs but by a desire for improved well-being and a higher quality of life. In addition, SH and VFR tourism are linked; in fact, as VFR tourists' stays are normally long, they are often housed, not in the homes of the relatives or friends that they are visiting, but in an SH—either their own or their host's or rented specifically [66,121].

SHs are also related to return migration, which concerns the return home of migrants who have been residents abroad for several years [13,17,20]. In the post-World War II period, many migrants believed they would return home after a certain period, and for this reason, they built or bought a home in their birthplace. Instead, as many were unable to actually return, the home built in the mother country has become, for many, an SH, whose use and maintenance has changed across generations [25,97,122,123].

While for the first generation the SH represented a place to return to, a 'home at home' [25] (p. 72), for subsequent generations it represents a tourist destination as well as a family home [25,69,124]. The subsequent generations sometimes buy an SH other than the original family home, displaying a greater connection to the broader place than with the house itself. Whereas for the first generation of emigrants, buying an SH in their place of origin symbolized social and economic success [125,126], for subsequent generations, the SH has become a tool to reconnect with their roots, strengthen their identity, learn more about the family homeland and its culture, and create or maintain relationships with distant relatives [59,104,127]. Sometimes, the second generation can give birth to flows of 'roots migration' or homeland returns [128,129], that is, 'migration to a place where members of the second generation originate from, but where they have never lived' [130] (p. 1084).

Contemporary SH owners tend to use them for increasingly diverse aims, ranging from leisure to seasonal to post-retirement use [26,75]. In some cases, having an SH can both trigger and constrain permanent migration after retirement. Possessing an SH may limit migration, because traveling to the SH is seen as a complement to staying in one's primary residence [131,132]. As such, owning an SH helps migrants enjoy two places in different moments, maximizing their quality of life and integrating urban and rural living in the process. Consequently, owning an SH may limit 'amenity migration,' a type of residential migration toward areas with high levels of natural amenities, such as mountains, seas, or good weather [133,134].

## 6. Conceptual Discussion

It is possible to show through an integrative framework the interrelationships among the variables synthesized in this study. The framework summarizes the findings of this analysis (see Figure 1). The figure illustrates the relationships described below.

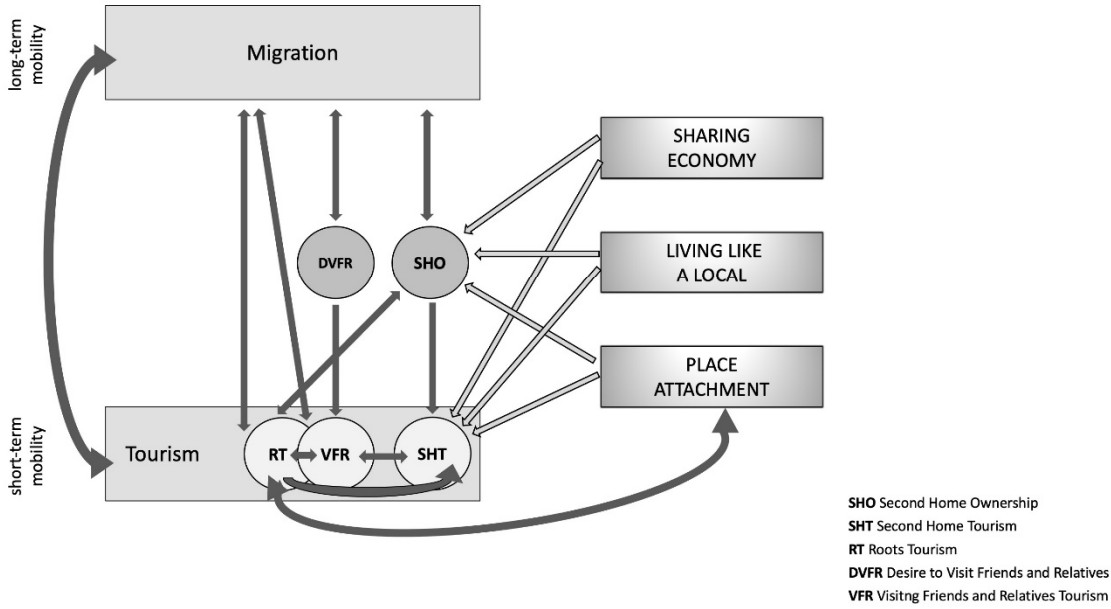

**Figure 1.** An integrative framework for understanding the interrelationships among different forms of tourism and migration. Source: Author's elaboration.

The two macro-variables of tourism and migration have a strong interrelation with each other (represented by the white arrow in the figure); there is a reciprocal influence between them that generates large flows of tourism-led migration and migration-led tourism.

There are two core variables in the tourism–migration relationship: the ownership of a second house (SHO) and the desire to visit friends and relatives (DVFR). An important aspect to be investigated is the effects that these variables may have as real or potential trigger elements linking tourism and migration. SHO is the most visible link between *temporary mobility* (tourism) and *permanent mobility* (migration): the owners of SHs are often considered residents rather than tourists, or at least 'permanent tourists.' SHO has a direct, bidirectional connection with migration. In fact, migrants buy SHs in their homeland, but frequently a substantial connection with a place where a person owns an SH is the motivation to choose it for future migration. In addition, often an SH is bought in view of a future migration, or it could be the motivation for the choice of a place as a new homeland. However, in some cases, SHO decreases the interest in emigrating. SHs could be the sources of flows of RT: roots tourists often wish to have a house in the land of their origins, or love visiting it when they have it. Of course, SHO also generates flows of SH tourism (SHT).

Another trigger element that creates a link between tourism and migration and can have a leverage effect is the desire to visit friends and relatives (DVFR in the figure). It may arise in the case of family and friends wishing to visit an emigrant and is the main motivator of VFR tourist segment. It can also give birth to other migration flows. VFR can be stimulated by migration but, as it has been explained above, the opposite is also true: People visit family members and friends who have emigrated, and these journeys can also be the basis of their own future migration choices.

When looking at other specific forms of migration-led tourism, is it possible to note that not only does migration generate RT, but tourism can give birth to emigration too. When people encounter new communities in different places during holidays, they can become interested in moving to live there. RT also has impact on VFR, sometimes being part of this larger tourist segment.

Moreover, it should not be overlooked that, as illustrated above, there are important reciprocal influences among the three forms of tourism considered (RT, VFR, and SHT).

Finally, in the figure are represented three relevant socio-cultural and economic phenomena: sharing economy, living like a local/home stay tourism, and place attachment. As explained in the analysis, they have current effects on the overall situation regarding tourism and migration but also have future potential influence on markets and consumer patterns. The sharing economy mainly has an impact on SHO and SHT, changing attitudes, preferences, and behaviours, especially in younger generations. In the meantime, phenomena, such as living like a local and home stay tourism, together with the spread of the sharing economy and especially accommodation sharing, are also changing lifestyles and preferences of tourists, increasing the interest towards local authenticity and SHs as tourist accommodation. In the future this can generate an important increase in SH markets but also, in the meantime, a minor interest in owing houses and other material assets. Finally, place attachment gives birth to a strong link with a site which influences different forms of tourism (SHT and RT) but also the choices regarding SHO and future migration choices.

*Limitations and Future Research Lines*

The research presented here is exploratory and, therefore, aims only to focus attention on a more comprehensive approach to the study of certain phenomena, including themes that are frequently studied separately, without offering final solutions. Analysis of the interrelationships between the various forms of tourism and migration identified requires further investigation. In particular, future studies should identify and carefully analyse the trends exhibited by the phenomena under examination in recent years and certain macro-variables (accommodation sharing, living like a local, lifestyle migration, etc.) that have influenced them. In addition, the practical implications of these trends for stakeholders and policy makers should be studied, so as to encourage the development of sustainable forms of tourism and limit the negative implications of some tourist activity.

As to future research lines, the results of this literature review highlight some aspects of the studied phenomena that should be investigated. In particular ways, there are evident trends in some socio-economic and cultural variables that could have an impact on them in the future. Certainly, the new forms of migration, high-skilled, lifestyle and so on, should be analysed with reference to the two macro-variables of SH and VFR tourism. In the future they will almost certainly become increasingly significant in Western countries and could have a major impact on some localities and their communities. Furthermore, the interactions between SHO, VFR and migration should also be studied with reference to some tourist segments that are strongly influenced by the tourists' family origins. Think, for example, of wedding tourism and edutourism [134]. Finally, in studying roots tourism and mobility the latest generations of descendants of migrants, who have extremely different preferences and behaviours compared to their predecessors, must not be neglected.

## 7. Conclusions

This study explores the effects of tourism on local area development through the links between VFR and SH tourism on the one hand and various types of migration that have emerged over time on the other. These links, which are simultaneously strong and multi-directional, have been understudied by researchers. The analysis presented shows that future research lines should be concentrated on the two key elements in the tourism–migration connection, DVFR and SHO, that have to date been little studied using such an approach. DVFR and SHO give birth to different forms of tourist consumption and, in the case of SHO, migration.

Tourism, mobility, and migration are international macro-phenomena that appear to be heavily interdependent. The most well-established connection between tourism and migration is SH tourism, with other forms of tourism, such as RT, closely related. Furthermore, SH ownership and SH travel, RT, and retirement, amenity, return, investment, and lifestyle migration are also interrelated, especially when the new place of residence is a

place of ancestral origin. In these cases, the SH, either purchased for holidays or inherited from the family, becomes a way of maintaining and strengthening tourists' roots, cultural identities, and family ties. In fact, emigrants and their descendants feel and share nostalgia. For them, the SH is a connection with their family history in the ancestral homeland and demonstrates their attachment and connection to it. In VFR tourism, the most frequent participants are migrants and their families and friends. In fact, VFR tourism gives rise to large flows of people traveling to visit each other after the migration of someone they know. Even if today ICTs allow you to meet family and friends more and more frequently on the internet, it is certainly not the same thing and it does not have the same value for people who feel the distance from their loved ones [135,136].

Many of the MLT segments described in this study reflect current and potential factors that may promote sustainable development, especially in secondary tourist destinations. For small, little-known and difficult-to-access villages, which are often affected by unemployment and depopulation, investments that seek to develop these forms of tourism locally may foster much-needed development by attracting visitors interested in discovering and experiencing local cultures, identities, and lifestyles. Future research should examine aspects of MLT related to sustainable development; in addition, the impact of these distinct types of tourism on destinations should be analysed with reference to the diverse dimensions of sustainability. In particular, factors related to the host–guest relationships deserve focused attention, as they represent a prerequisite for the success of any form of development driven by tourism.

**Funding:** This research received no external funding.

**Institutional Review Board Statement:** Not applicable.

**Informed Consent Statement:** Not applicable.

**Data Availability Statement:** Not applicable.

**Conflicts of Interest:** The author declares no conflict of interest.

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
