# Peer review of "Impacts of Second Home and Visiting Friends and Relatives Tourism on Migration: A Conceptual Framework"

_sustainability, doi:10.3390/su14074352_

Round 1
Reviewer 1 Report
Dear authors,
Thank you for submitting this manuscript. I found it very interesting, however I have a few suggestions to improve the overall quality of the paper.
Introduction can be shorter focusing on the main issue and significance of the study. The rest can go to the literature review section. The author should also pay a better attention to the contributions of phenomena such as roots tourism to sustainability such as long-lasting connections with the destination and wellbeing of travellers.
You should add a method section to explain what process was followed. Was it systematic review or what sources, how many were used?
English editing is required.
Author Response
Good evening. I thank you very much for your help. According to your kind suggestions I made the followings changes:
I made the introduction shorter and moved many concepts in the literary review section.
I inserted something about roots tourism sustainability (lines 230-235).
I explained better the methodology (lines 80-88).
English editing was done.
Reviewer 2 Report
I enjoyed reading your paper. There is a lot of value to the work you are presenting. However there are a few suggestions for improvement. These are listed below
- I would suggest a re-wording of the title to include conceptual framework. The title is also wordy and does not succinctly state what the work presented is about.
- Paragraphs 1-3 of the introduction needs revisiting to ensure it connects with the rest of the paper. When I read these I was left wondering how do all of these things link together based on the title and the abstract. It is only when I got to paragraph 4, I got it. Paragraphs 1-3 should be used to make a stronger case for the research.
- As this is a conceptual paper you should make it clearer the process you went through to develop the conceptual thinking behind the paper. In other words, a short methodology section. You paper reads like a very well constructed narrative literature review but you need to strengthen it beyond it to show the development of the conceptual framework.
- I liked that you explained the conceptual framework but to make this paper stronger you need to be clearer about future research directions and the contribution of your work to the field. How does progressing knowledge about the links between SH and VFR add to the body of knowledge? For me, you need to draw out a section on theoretical contribution and what are the future directions for the field. Some of what you have written can fall into diaspora tourism.
Reviewer 3 Report
I have read the article with due attention and interest.
The article is conceptual but based on a literature review, therefore, there is explanation is needed of the "searching and selection" of literature methodology. In the lines 104-106 author(s) wrote „Moreover, it may encourage an innovative approach to the study of phenomena that have been studied but which—in recent decades—have been changing rapidly, and therefore need different analysis tools and a new research perspective". In the article this needs to be describe clearly, for example what analysis tools. Good to think from the perspective, what are the new and useful findings that we want to share here,
Please note, that in information society era and ICT development people may visit someone in real whom meet in social media, and it is not related to migration (line 373) and such situations will become more and more frequent.
The abstract must be rephrased – it should include: the purpose of the article, the research methods and the research findings.
The transparency of the figure should be improved - the reviewer is presented with a black and white figure, white the autor(s) writes about a red arrow (line 297).
Author Response
Good evening. I thank you very much for your help. According to your kind suggestions, I made the followings changes in the paper:
I explained better the methodology (lines 80-88) and delete the part where I have written about the need for different analysis tools.
I wrote something about the importance of meeting friends and relatives face to face (lines 395-397).
I inserted in the abstract the aim, the research methodology and the findings.
I changed the figure. Now the arrow is white.
Round 2
Reviewer 1 Report
Thank you for the revisions, however the manuscript still lacks a method section explaining data collection and analysis in detail. The method used (qualitative systematic literature review) must be explained for example, what data base, year, number of papers and how it was analysed? What was the process?
Author Response
Dear reviewer,
I thank you very much for your help. As suggested by you, I tried to explain the different phases and the methodology of the research process in a paragraph.
Reviewer 2 Report
I commend the author for taking the time to make the required changes. This paper reads more comprehensive now. There are a few typos. Please correct.
Author Response
Dear reviewer,
I thank you for your help. I corrected the typos.
Reviewer 3 Report
It has not been explained still what is meant by an innovative approach to the study (line 410) and how the literature was searched for a qualitative systematic literature review.
Author Response
Dear reviewer,
I thank you very much for your help. As suggested by you, I tried to explain the different phases and the methodology of the research process in a paragraph.
As to the innovative approach to the study, more than that was thinking about a more extensive and comprehensive study, including subthems that frequently are studied separately. I explained it.
Round 3
Reviewer 1 Report
The manuscript can be accepted.
Reviewer 3 Report
The introduced corrections increased the substantive level of the article and in this form it can be published.